# GA-Based Optimization Method for Mobile Crane Repositioning Route Planning

**Han-Seong Gwak [1,*], Hong-Chul Lee [2], Byoung-Yoon Choi [3] and Yirong Mi [3]**

1   Construction Engineer Policy Institute of Korea, Seoul 06098, Korea
2   Intelligent Construction Automation Center, Kyungpook National University, Daegu 41566, Korea; colf@knu.ac.kr
3   School of Architecture, Environmental, Energy and Civil Engineering, Kyungpook National University, Daegu 41566, Korea; jr1381@knu.ac.kr (B.-Y.C.); 2021320938@knu.ac.kr (Y.M.)
*   Correspondence: hsgwak@cepik.re.kr; Tel.: +82-2-6240-4336

**Abstract:** Mobile cranes have been used extensively as essential equipment at construction sites. The productivity improvement of the mobile crane affects the overall productivity of the construction project. Hence, various studies have been conducted regarding mobile crane operation planning. However, studies on solving RCP (the repositioning mobile crane problem) are insufficient. This article presents a mobile crane reposition route planning optimization method (RPOS) that minimizes the total operating time of mobile crane. It converts the construction site into a mathematical model, determines feasible locations of the mobile crane, and identifies near-global optimal solution (s) (i.e., the placement point sequences of mobile crane) by implementing genetic algorithm and dijkstra's algorithm. The study is of value to practitioners because RPOS provides an easy-to-use computerized tool that reduces the lengthy computations relative to data processing and Genetic Algorithms (GAs). Test cases verify the validity of the computational method.

**Keywords:** mobile crane; reposition; genetic algorithm; optimization; data modeling

## 1. Introduction

Cranes are construction equipment that are used to lift loads horizontally or vertically [1]. Cranes have been used extensively as essential equipment in construction because heavy objects are handled frequently at construction sites. Furthermore, as the demand for large-scale construction projects has increased and the benefits of modular construction have been recognized, the use of mobile cranes for lifting and installing large modular materials has also been increasing [2]. Mobile cranes have a built-in motor, which allows it to move to a specific location independently. This makes mobile cranes easier to install, dismantle, and relocate for lifting work compared to tower cranes.

In modular construction, the mobile crane repeats the task of hanging a load on the hook and transporting it to the installation location. This requires planning, such as selecting an appropriate crane and choosing the crane position while considering productivity and safety. A careful lift plan is required to perform the lifting work successfully. An improper lift plan may lead to high additional costs and schedule delays [2]. Hence, various studies have been conducted to establish optimal lift plans. These studies can be classified into three research themes: (1) optimal mobile crane selection [3–6], (2) investigation on feasible working areas for mobile cranes [7,8], and (3) lift path optimization of mobile cranes [9–15]. As such, many studies have been conducted on mobile crane operation plans. However, very few studies have been conducted to investigate the optimal mobile crane position and travel route.

A mobile crane can lift loads within the working range determined by the boom length, vehicle size, work safety regulations, etc. [7,13]. Generally, because the area of a large site is wider than the working range of the mobile crane, the crane cannot be fixed at

a certain position and lift loads all over the site. However, it may not be economical to hire and place multiple mobile cranes all over the site. The construction site has a construction schedule according to the construction plan, and the movement of the mobile crane to the appropriate position must be planned to lift loads according to the construction schedule and workload. In this study, we named this problem as the repositioning mobile crane problem (RCP).

Traditionally, RCP was planned by relying on the intuition and experience of the construction manager or workers. However, to solve RCP, various parameters must be considered simultaneously: the three-dimensional geometry of the construction structure, volume, and shape of the crane, working motions and range of the crane, performance of the crane, workload in each zone, weight of the load, etc. Furthermore, because RCP has a very large solution space, it is difficult to solve based on intuition, and an improper plan can be established easily. In other words, RCP should be solved using a scientific approach, and the development of an integrated system that combines various techniques is required.

In this study, we propose a method for solving RCP with minimal relocations and in the shortest period. Based on this method, a system named Mobile Crane Reposition route Planning Optimization System (RPOS) was developed. To achieve the goal of this study, the following specific research procedures were performed. First, prior studies on mobile crane operation plans were analyzed to determine the need for a new method to achieve academic and practical progress. Second, the construction site, crane performance, and lift constraints were schematized to model them using numerical data. Third, a genetic algorithm (GA) was used to develop RPOS. Fourth, a case study was conducted to validate the detailed application methods and performance of RPOS.

## 2. Current State of Mobile Crane Operation Planning

Many studies have been conducted to establish mobile crane operation plans. Crane operation planning starts with the selection of an appropriate crane by considering the site conditions. Existing studies have identified essential factors that need to be considered for the selection of an appropriate crane (i.e., work clearance, lift height, lift performance of the crane, ground condition, site accessibility, etc.) Considering these conditions, Hanna and Lotfallah [3] used fuzzy logic to select the crane type. Sawhney and Mund [4] developed a stochastic artificial neural network (ANN)-based crane type selection system.

Wu et al. [5] developed an algorithm for selecting an appropriate crane by considering the geometric shape based on the payload and boom/jib of the crane. Hasan et al. [16] argued that the wind direction is crucial for crane selection owing to the characteristics of the lift work and proposed a building information modeling (BIM)-based crane selection method. Thus, based on existing studies, the optimal crane type can be selected by considering various site conditions.

Once the mobile crane selection is completed, feasible locations where the crane can be installed/placed to operate on the site are searched. As the feasible locations must be considered according to the crane type and site conditions while selecting the crane, a study has been conducted to select the crane and feasible locations simultaneously [17]. The method proposed by Al-Hussein finds feasible working areas for the crane based on the lift performance and maximum lift range of a particular crane. Tantisevi and Akinci [8] identified the work areas that minimized the potential spatial conflicts between the crane and the structure. Safouhi et al. [7] automated the feasible working area search procedure for a crane by considering the minimum and maximum working ranges of the crane and the rotational range of the vehicle body in a computer-aided design environment. Furthermore, Wang et al. [18] proposed a method for finding a work area to increase the collaborative productivity of cranes when multiple cranes are operated on a site. Based on the outcomes of the existing studies, workers can determine the feasible working areas of the crane by considering the lift performance of the crane, structure, three-dimensional geometry of the crane, shape, and weight of the load, etc. However, existing studies have focused on lift locations and load characteristics. Thus, there is a lack of consideration for loading areas.

Owing to the nature of the construction site, materials must be loaded in the designated places. Therefore, for feasible working areas of a crane, the loading areas must be included in the working range of the crane. Therefore, a method to find feasible working areas by considering the loading locations, lift locations, and working range of the crane is required.

Once the feasible working areas are identified, the crane can be positioned at an appropriate location to lift loads. However, the manager must review the adequacy of the lift path because the lift path is directly linked to productivity and occurrence of safety accidents. Sivakumar et al. [9] used the hill-climbing and A* algorithm to propose a method that searches for the optimal lift path when two cranes collaborate to lift a load. Ali et al. [10] investigated the GA-based optimal lift path. Kang and Miranda [11] considered the shape of the payload and lifting behavior to search the lift path. Zhang and Hammad [19] proposed an algorithm to find a lift path by considering the smoothness of the crane motion. Moreover, Cai et al. [20] proposed a GA-based method for finding a lift path that can increase the time and energy efficiencies by considering the characteristics of the crane operator.

Numerous studies have been conducted on three research themes regarding mobile crane operation planning (crane selection, feasible working area identification, and lift path search for the crane). However, as mentioned earlier, studies on solving RCP are insufficient. Only some studies have mentioned the RCP. Lei et al. [21] proposed an algorithm that finds the walking path of the mobile crane. The algorithm is similar to the method for solving RCP as it considers the movement of the crane, unlike previous studies. However, Lei [15] defined the walking path as the path required for a crane to move, while the load is suspended from the hook to a location where lifting can be performed easily. However, the method proposed by Lei cannot be used for solving RCP because it does not search an installation location for the mobile crane according to the construction sequential order. Studies directly involved in solving the RCP were conducted by Tantisevi and Akinci [8] and Pan et al. [22]. Both studies mentioned the importance of travel route planning of the mobile crane because relocating a mobile crane is time-consuming. Tantisevi and Akinci [8] developed an algorithm that minimizes the frequency of relocating the mobile crane according to the construction sequential order and working range of the crane. On the other hand, Pan et al. [22] developed an algorithm that finds the optimal walking path by identifying the coincident point of feasible locations according to the lift locations and comparing the working speed of the crane. However, existing studies on solving RCP have the following limitations. First, existing studies suggested heuristic-based methods. The heuristic method is widely used because it is easy to understand, and the computational speed is fast. However, it cannot guarantee the accuracy of the computational results. An improper crane operation plan can incur additional costs and schedule delays. Therefore, heuristic-based methods are not suitable for crane travel route planning, and a global solution must be found. Second, the realistic constraints of RCP and their accompanying variabilities of crane productivity have not been presented mathematically. Third, when a mobile crane is relocated from a particular position to another, the moving constraints are not considered. Obviously, construction sites have sections that are inaccessible to cranes with large volumes. Therefore, when a crane has to move from a particular position to another, the travel routes should be found quickly by excluding the inaccessible sections, and the travel time for each route should be estimated and used as a criterion in the travel route planning of the crane. Fourth, an integrated system that can increase the practical applicability has not been provided. In this study, we propose a method that overcomes the limitations observed in existing RCP studies while facilitating optimal mobile crane travel route planning.

## 3. Mobile Crane Repositioning Route Optimization Method

RPOS searches for a reposition route plan that minimizes the mobile crane operating time. All computational processes were implemented using MATLAB R2014b-based software. RPOS comprises of four modules: (1) site data modeling, (2) identifying feasible

locations for the mobile crane, (3) GA-based near-global optimal reposition route planning for the mobile crane, and (4) outputting the near-global optimal mobile crane reposition plan and operating time report. A detailed description of each module is provided below.

### 3.1. Site Data Modeling

The mobile crane installation is limited owing to work safety issues and the physical attributes of the crane. The manager must determine whether the road surface is adequately strong to install outriggers to ensure the safety of mobile crane operation. Additionally, the manager must consider whether there are obstacles, such as high-voltage power lines, telephone poles, streetlights, and roadside trees. Furthermore, the working range ($R_W$) is defined by the physical attributes (boom length, vehicle size, etc.) of the mobile crane and the work safety regulations. There are multiple demand points ($S_L = \{S^1, S^2, ..., S^i\}$) and supply points ($P_L = \{P^1, P^2, ..., P^j\}$) on the site, and their locations and lifting radius should be considered. The RPOS determines the feasible locations by considering (1) points already occupied by buildings, etc., (2) supply points, (3) points where the bearing capacity of soil has not been secured, (4) points where the safe work distance has not been secured, and (5) lifting radius of the mobile crane. Figure 1 shows the method and procedure of RPOS for identifying the feasible locations of the mobile crane in a sample field.

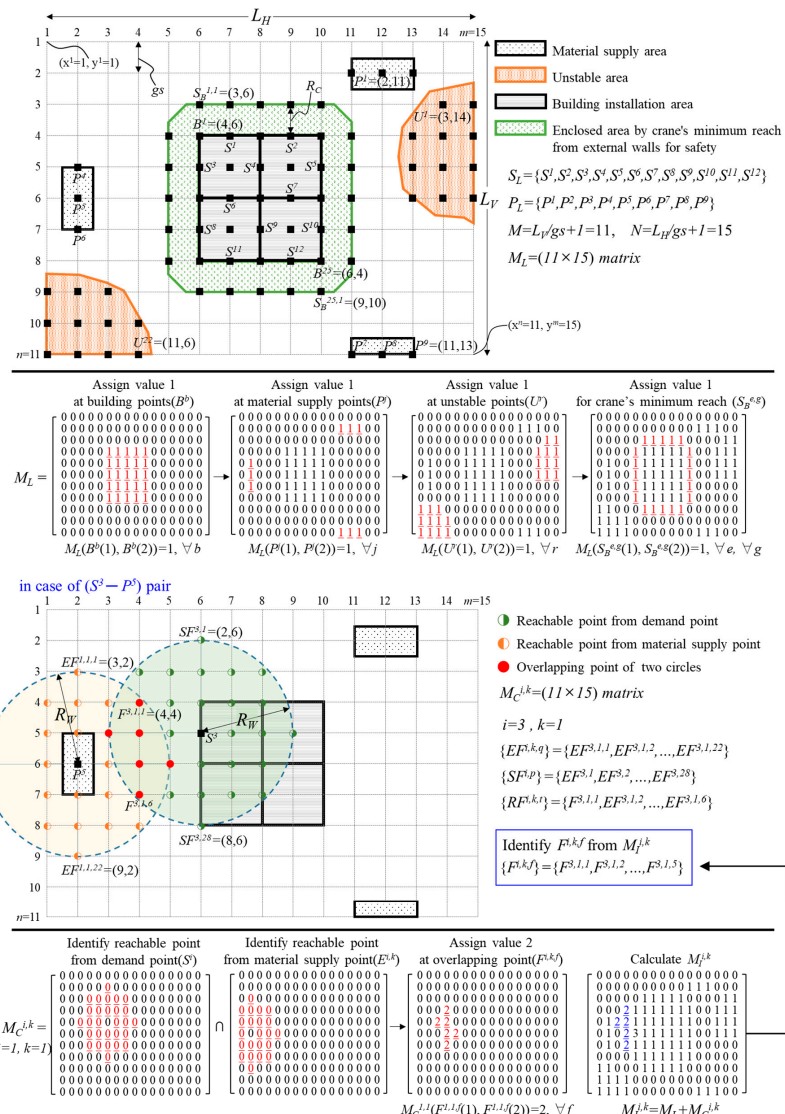

**Figure 1.** Procedure of identifying the feasible locations of the mobile crane.

First, RPOS divides the site using the X and Y axes and sets the upper leftmost point of the site as the reference point ($x1 = 1$, $y1 = 1$). This is done to synchronize the site grid coordinate system and system matrix structure. The grid size ($gs$) affects the precision and computational speed of finding the optimal solution. In other words, as $gs$ decreases, the practicality of the mathematical model increases, but the computational time for finding the optimal solution increases. The user considers this relation when determining $gs$. If $gs$ has been determined, the RPOS converts the site into an $M \times N$ orthogonal coordinate system. Furthermore, a matrix $M_L$ of $M \times N$ size that has 0 as the attribute value is created: $M_L = $ zeros ($M, N$). $M_L$ is a matrix for storing points in the system memory, which are identified as locations where the mobile crane cannot be installed. Here, $M$ and $N$ are determined by dividing the vertical length ($L_V$) and horizontal length ($L_H$) of the site area, respectively, by $gs$ (Equation (1)).

$$M = \frac{L_V}{gs} + 1, \quad N = \frac{L_H}{gs} + 1. \tag{1}$$

Next, RPOS defines the coordinates of areas occupied by buildings. For this, the Industry Foundation Classes (IFC) file extracted from the BIM model is input into the system. The IFC file is a text file containing entity instances according to the IFC EXPRESS schema. The location information of the building components is managed by the IfcProduct entity. RPOS synchronizes the coordinate system used by IfcProduct and the on-site grid coordinate system by performing the origin coordinate correction process.

### 3.2. Feasible Location Identification for the Mobile Crane

RPOS identifies the points occupied by building ($B^b$) and assigns a value of 1 to the positions corresponding to the (x, y) coordinates of $B^b$ among the $M_L$ components (Equation (2)). Mobile crane cannot be installed at the supply points ($P^j$). RPOS assigns a value of 1 to the position corresponding to the (x, y) coordinate of $P^j$ among the $M_L$ components (Equation (3)).

$$M_L\left(B^b(1), B^b(2)\right) = 1, \; \forall b, \tag{2}$$

$$M_L\left(P^j(1), P^j(2)\right) = 1, \; \forall j. \tag{3}$$

The user selects an appropriate mobile crane by considering the site conditions and inputs the data into the system. RPOS queries the physical attribute data of the user-selected mobile crane from the equipment database and stores them in $C_I$, as shown in Equation (4).

$$C_I = (W_C, R_C, L_B, H_C, H_R), \tag{4}$$

where $W_C$ is the weight of the mobile crane, $R_C$ is the minimum radius for safe lifting, $L_B$ is the maximum boom length, $H_C$ is the height of the mobile crane body, and $H_R$ is the minimum rope length.

Next, RPOS considers the weight of the selected mobile crane ($W_C$), geotechnical investigation report, and GIS data to determine whether the outriggers should be installed on the ground and identifies the points ($U^r$) where the mobile crane cannot be installed. RPOS assigns a value of 1 to the position corresponding to the (x, y) coordinate of $U^r$ among the $M_L$ components (Equation (5)).

$$M_L(U^r(1), U^r(2)) = 1, \; \forall r. \tag{5}$$

For safe lifting work, the mobile crane must maintain a certain distance from the facilities that hinder the lifting work, such as the outer walls of buildings. This is called the minimum radius of the mobile crane ($R_C$). RPOS extracts the points ($E^e$) corresponding to the outer walls of the buildings among the points ($B^b$) occupied by the building components

($E_B = \{E^1, E^2, ..., E^e\}$). If it is determined that a point adjacent to the outer wall of a building is located within $R_C$, then it is excluded from the feasible locations of the mobile crane.

RPOS identifies the points ($S_B{}^{e,g}$) where $R_C$ is not secured from each outer wall point ($E^e$) of the building (Equation (6)) and assigns a value of 1 to the position corresponding to the (x, y) coordinate of $S_B{}^{e,g}$ among the $M_L$ components (Equation (7)).

$$S_B{}^{e,g} = N_D{}^n; \ \sqrt{(E^e(1) - n(1))^2 + (E^e(2) - n(2))^2} \leq \frac{R_C}{gs}, \ \forall n, \tag{6}$$

where $e$ is the index of the outer wall point ($E^e$), $E^e(1)$ and $E^e(2)$ are the x and y coordinates of the $e$-th outer wall point, respectively, $n$ is the index of the on-site grid coordinate point, and $N_D{}^n$ is on-site grid coordinate point.

$$M_L(S_B{}^{e,g}(1), S_B{}^{e,g}(2)) = 1, \ \forall e, \ \forall g. \tag{7}$$

RPOS calculates the maximum radius ($R_W$) of the mobile crane. $R_W$ is calculated using Equation (8) by considering not only the physical characteristics of the mobile crane but also the length of the payload ($L_M$) and height of demand point ($H_I$) simultaneously.

$$R_W = \sqrt{L_B{}^2 - (H_I - H_B + L_M + H_R)^2}, \tag{8}$$

where $L_B$, $H_B$, and $H_R$ call the corresponding values from $C_I$ defined in Equation (4), and $L_M$ is defined by the user. Furthermore, $H_I$, which is called from the IfcProduct coordinate system, is the height information (z value) of a particular load demand point ($S^i$).

Lifting is performed by pairing a demand point and a supply point. The demand and supply points must be located within the working range of the mobile crane ($R_W$); otherwise, the lifting work is impossible. However, the demand and supply points may be in the one-to-many relationships. For example, at demand point I1, as shown in Figure 2, loads can be lifted from two supply points (L1 and L2). RPOS identifies the available supply points ($E^{i,k}$) where loads can be lifted for each demand point ($S^i$) based on $R_W$ (Equation (9)).

$$E^{i,k} = P^j; \ \sqrt{(S^i(1) - P^j(1))^2 + (S^i(2) - P^j(2))^2} \leq \frac{2 \times R_C}{gs}, \ \forall j, \tag{9}$$

where $i$ is the index of the demand point ($S^i$), $S^i(1)$ and $S^i(2)$ are the x and y coordinates of the $i$-th demand point ($S^i$), respectively, $j$ is the index of the supply point ($P^j$), $P^j(1)$ and $P^j(2)$ are the x and y coordinates of the $j$-th supply point ($P^j$), respectively.

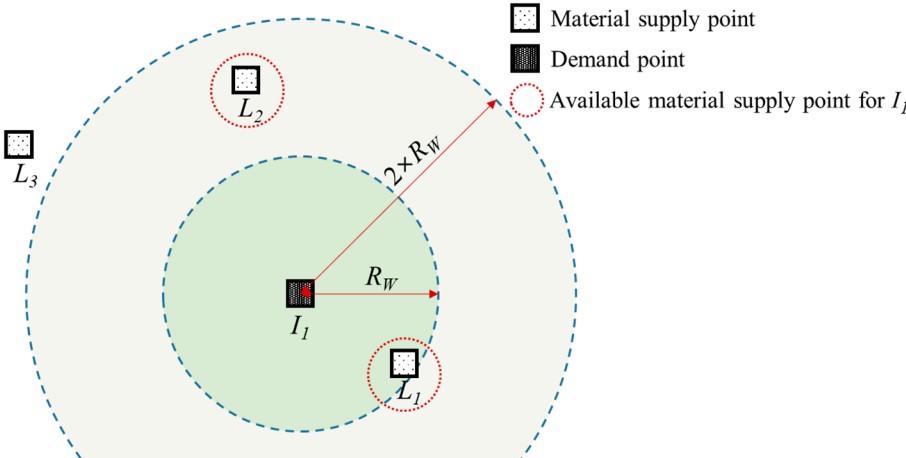

**Figure 2.** Working range of the mobile crane and the available supply point for the demand point.

Next, RPOS creates a matrix $M_C{}^{i,k}$ to store the feasible points of the mobile crane for each pair of demand point ($S^i$) and supply point ($E^{i,k}$). $M_C{}^{i,k}$ is a matrix of $M \times N$ size and has 0 as an attribute value: $M_C{}^{i,k}$ = zeros ($M$, $N$).

RPOS identifies the coincident points ($EF^{i,k,q}$) for the $S^i - E^{i,k}$ pair by intersecting the sets of points ($SF^{i,p}$) located within the range of $R_W$ centering on the $S^i$ and points ($RF^{i,k,q}$) located within the range of $R_W$ centering on the $E^{i,k}$ (Equation (10)). RPOS assigns a value of 2 to the component corresponding to the (x, y) coordinate of $RF^{i,k,t}$ among the components of $M_C{}^{i,k}$ (Equation (11)).

$$\left\{ RF^{i,k,t} \right\} = \left\{ SF^{i,p} \right\} \cap \left\{ EF^{i,k,q} \right\}, \ \forall p, \ \forall q$$
$$s.t.$$
$$SF^{i,p} = n; \ \sqrt{\left(S^i(1) - n(1)\right)^2 + \left(S^i(2) - n(2)\right)^2} \leq \frac{R_W}{gs}, \ \forall n \qquad (10)$$
$$EF^{i,k,q} = n; \ \sqrt{\left(E^{i,k}(1) - n(1)\right)^2 + \left(E^{i,k}(2) - n(2)\right)^2} \leq \frac{R_W}{gs}, \ \forall n$$

$$M_C{}^{i,k}\left( RF^{i,k,t}(1), RF^{i,k,t}(2) \right) = 2, \ \forall t, \qquad (11)$$

where $n$ is the index of the on-site grid coordinate point, $p$ is the index of the point ($SF^{i,p}$) located within the range of $R_W$ centering on the demand point $S^i$, and $q$ is the index of the point ($EF^{i,k,q}$) located within the range of $R_W$ centering on the supply point $E^{i,k}$.

RPOS performs the addition of $M_L$ and $M_C{}^{i,k}$ to derive $M_I{}^{i,k}$ (i.e., $M_I{}^{i,k} = M_L + M_C{}^{i,k}$). Additionally, RPOS determines the set of points where the attribute value of $M_I{}^{i,k}$ is 2 ($F_L{}^{i,k}$ = {$F^{i,k,1}$, $F^{i,k,2}$, ..., $F^{i,k,f}$}) as feasible points ($F^{i,k,f}$) for the $S^i - E^{i,k}$ pair. The points where the attribute value of $M_I{}^{i,k}$ is 1 or 3 indicate the points already occupied by buildings, etc., ($B_b$), points where the bearing capacity of soil is not secured ($R^k$), or points where the safe work distance is not secured ($S_B{}^{e,g}$). $F^{i,k,f}$ for all demand points ({$S^1$, $S^2$, ...,$S^i$}) is stored in the system memory.

### 3.3. Near-Global Optimal Mobile Crane Reposition Route

Meta-heuristics such as GA, SA, and TS are effective in solving the problem in this study. All three meta-heuristics have the ability to find better solution than the manual process. Several studies on the performance comparison of GA, SA, and TS have been conducted [23–25]. Lidbe et al. [23] show that TS give better calibration results compared to the GA and SA. On the other hand, the study by Said et al. [25] found that GA has a better solution quality than SA, TS for solving QAP (Quadratic Assignment Problem) optimization problems, but, TS has a faster execution time than the others. This suggests that there may be differences in the performance of GA, SA, and TS depending on the type of problem. Therefore, this study adopts GA, which is the most diverse and widely used in the construction field, to solve the RCP problem.

This module uses GA to find the near-global optimal reposition route for the mobile crane. The chromosome type, fitness function, and GA operation termination rules used to find the near-global optimal reposition route of the mobile crane are provided in detail below.

First, the GA parameters (i.e., population size ($PS$), crossover rate ($CR$), mutation rate ($MR$)) are set by the user. Then, RPOS sets the GA operation termination rules. For the termination rules, we used the four termination criteria provided by MATLAB: (1) Generation: the operation is terminated if the maximum number of generations is reached; (2) Stall-GenLimit: the operation is terminated if there is no improvement in the objective function during a specific number of generations (stall generation); (3) StallTimeLimit: the operation is terminated if there is no improvement in the objective function for a certain period; (4) TimeLimit: the operation is terminated if the maximum computation time is exceeded. Tolfun (= $1 \times 10^{-6}$), which is set as a default value by the GA solver of MATLAB, is used to determine whether the objective function has improved.

GA chromosomes are composed of a set of genes (*gn*). In RPOS, the GA chromosome is composed of a pair of the index (*k*) of the supply point ($E^{i,k}$), where the load can be lifted at a certain demand point ($S^i$), and the index (*f*) of the feasible point ($F^{i,k,f}$) for the $S^i - E^{i,k}$ pair (Figure 3).

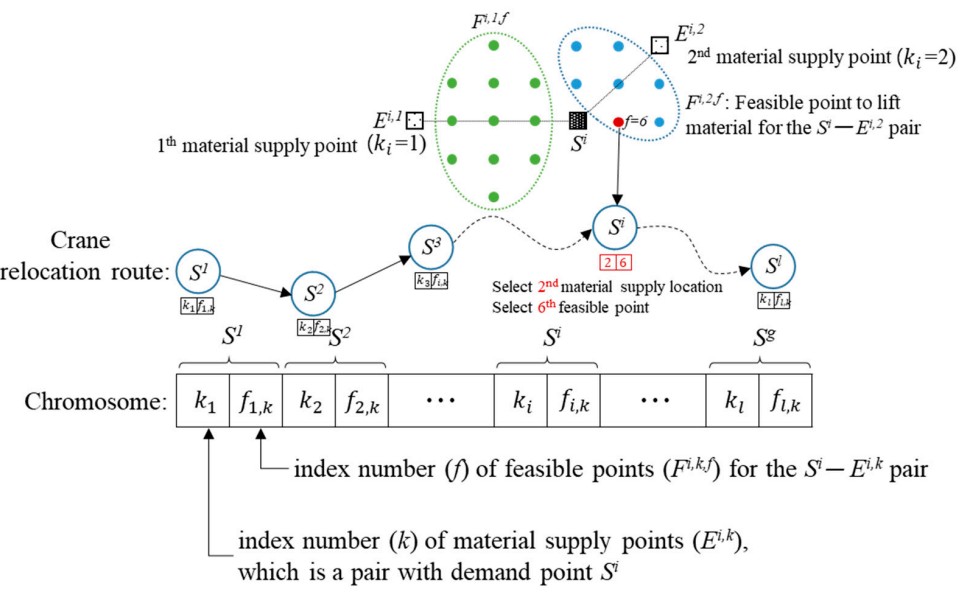

**Figure 3.** Encoding a chromosome.

The serial number of a gene (*gn*) pair is the same as the index (*i*) of the demand point. This number is determined by the load installation sequential order according to the construction plan. In other words, after installing the *i*-th load, the *i* + 1 load was installed. The gene serial number also follows this sequential order. The chromosome length, which is twice the number of loads to be installed (*l*), is determined by the number of installing loads. The search range of the optimal solution ($S_S$) is the number of combinations of the supply points ($E^{i,k}$) where loading is feasible at the demand point ($S^i$) and the feasible points ($F^{i,k,f}$) for the $S^i - E^{i,k}$ pair (Equation (12)).

$$S_S = \prod_i \left( \sum_{i,k}^{n(E^{i,k})} n\left( F^{i,k,f} \right) \right). \tag{12}$$

RPOS sets the objective function of the GA as the total operating time of the crane to search for the optimal reposition route of the mobile crane. The total operating time is calculated by adding the load lifting time (*ht*), crane traveling time (*tt*), and crane installing/dismantling time (*rt*). As the optimal reposition route minimizes the total operating time, the fitness function $f_{OT}$ is set as the inverse number of the total operating time (Equation (13)).

$$f_{OT} = \frac{1}{\sum_{i=1}^l ht(k_i, f_{i,k}) + \sum_{i=2}^l tt(f_{i-1,k}, f_{i,k}) + \sum_{i=2}^l rt \times a_i}, \tag{13}$$

where $ht(k_i, f_{i,k})$ is the time required when the mobile crane is positioned at a point $f_{i,k}$-th feasible point for the $S^i - E^{i,k}$ pair and lifts the load of the supply point $E^{i,k}$ to the demand point $S^i$; $tt(f_{i-1,k}, f_{i,k})$ is the time required when the crane moves from $i-1$th crane placement point to *i*-th placement point; rt is the time required to install/dismantle the mobile crane; $a_i$ is a binary number that determines whether to install or dismantle the mobile crane.

$ht(k_i, f_{i,k})$ is calculated using the lifting time estimation function of the crane defined using an ANN by Tam et al. [26] (Equation (14)). The input variables of the lifting time

estimation function ($f_{hoistt}$) are the weight of the load ($W_M$), rotation angle of the crane ($A_C$), distance between the supply point and demand point ($D_{LI}$), and height ($H_I$) of the demand point. The rotation angle of the mobile crane ($A_C{}^i$) and distance between the supply point ($E^{i,k}$) and demand point ($S^i$) are calculated using Equations (15) and (16), respectively. The height ($H_I{}^i$) of the supply point ($S^i$) is called from the IfcProduct coordinate system. The weight of the load ($W_M$) is defined by the user.

$$ht(k_i, f_{i,k}) = f_{hoistt}\left(W_M, A_C{}^i, D_{LI}{}^i, H_I{}^i\right), \tag{14}$$

$$AC^i = atan\left(\frac{E^{i,k}(2) - f_{i,k}(2)}{E^{i,k}(1) - f_{i,k}(1)}\right) - atan\left(\frac{S^i(2) - f_{i,k}(2)}{S^i(1) - f_{i,k}(1)}\right), \tag{15}$$

$$D_{LI}{}^i = \sqrt{\left(E^{i,k}(1) - S^i(1)\right)^2 + \left(E^{i,k}(2) - S^i(2)\right)^2}, \tag{16}$$

RPOS uses Dijkstra's algorithm (DA) to calculate the traveling time $tt(f_{i-1,k}, f_{i,k})$ between the points of the mobile crane. DA is an algorithm used for finding the shortest paths from a starting point to all other points. The DA operation requires a two-dimensional array called a weighted adjacency matrix. Weights can be interpreted in various ways, including cost, distance, and time. In this study, the weight is the traveling time of the mobile crane. The weighted adjacency matrix ($G$) was constructed by setting the traveling time value by considering the link length, traveling performance of the crane, and geotechnical attributes (i.e., rolling resistance, grade resistance, etc.) if a link existed between two nodes or by setting an infinite value if there was no link. After constructing the weighted adjacency matrix (G), the traveling time from $i$–1th crane placement point to $i$-th placement point was calculated using the distances function (Equation (17)). However, using DA to calculate the point-to-point traveling time is valid assuming that the crane operates on the shortest distance when traveling between two points.

$$tt(f_{i-1,k}, f_{i,k}) = distances(G, f_{i-1,k}, f_{i,k}). \tag{17}$$

The time $rt$ required for installing/dismantling the mobile crane is defined by the user. The mobile crane is installed in a series of processes, including pedestal installation, outrigger deployment, and horizontal balance inspection. On the other hand, the crane is dismantled by the processes of outrigger retraction, pedestal removal, etc. The user considers the mobile crane type, installation/dismantling procedures, and proficiency of the operator to determine an appropriate $rt$ value and input it into the system. $a^i$ is a binary number that determines whether to install/dismantle the crane; and if the $i$-th crane placement point ($F^{i,k,f}$) is the same as the previous ($i - 1$) point ($F^{i-1,k,f}$), then 1 is assigned; if not, 0 is assigned (Equation (18)).

$$a_i = \left\{ \begin{array}{l} 1, \ \ if \ F^{i,k,f} == F^{i-1,k,f} \\ 0, \ \ otherwise \end{array} \right. . \tag{18}$$

The GA generates the initial population based on *PS* and gradually improves the chromosomes via numerous evolutionary generations to converge to the near-global optimal solution. The superior and inferior chromosomes of each generation are distinguished by the fitness function ($f_{OT}$), and the superior chromosomes are selected with high probabilities. Eventually, the inferior chromosomes are removed as the generation progresses, and the process of mutating (mutation) or exchanging (crossover) the genetic information between the superior chromosomes is repeated. The GA iterations stop as soon as any of the stopping rules are satisfied. The fitness function identifies a near-global optimal set of pairs of the mobile crane location point and material supply points for each demand point that minimizes the total operating time.

### 3.4. Output of Near-Global Optimal Reposition Route and Operating Time Report

After completing the optimization operation of the GA, RPOS provides the user with visual information, such as (1) the placement point sequences of the mobile crane, (2) demand point-supply point pair, (3) travel route between the position points of the mobile crane, and (4) total crane operating time.

## 4. Method Verification

### 4.1. Overview of Case Study Site and Site Data Modeling

To validate the effectiveness of RPOS, we constructed a test site, as shown below, and developed a BIM model (Figure 4). At the test site, a precast concrete construction method was used for the structural work, and 15 columns were erected according to the construction sequences. The column was 3.5 m long and weighed 0.7 tons. The column installation sequence was same as the number indicated at the column installation points shown in Figure 4. There were 10 places on the site where the columns could be loaded. The site dimensions were 80 × 60 m. Additionally, there were places on the site where the mobile crane could not move to. The mobile crane used in the test was assumed to have a maximum boom length ($R_W$) of 16 m, minimum radius for safe lifting ($R_C$) of 5 m, a traveling speed of 1 m/s, and an installation/dismantling time of 10 min.

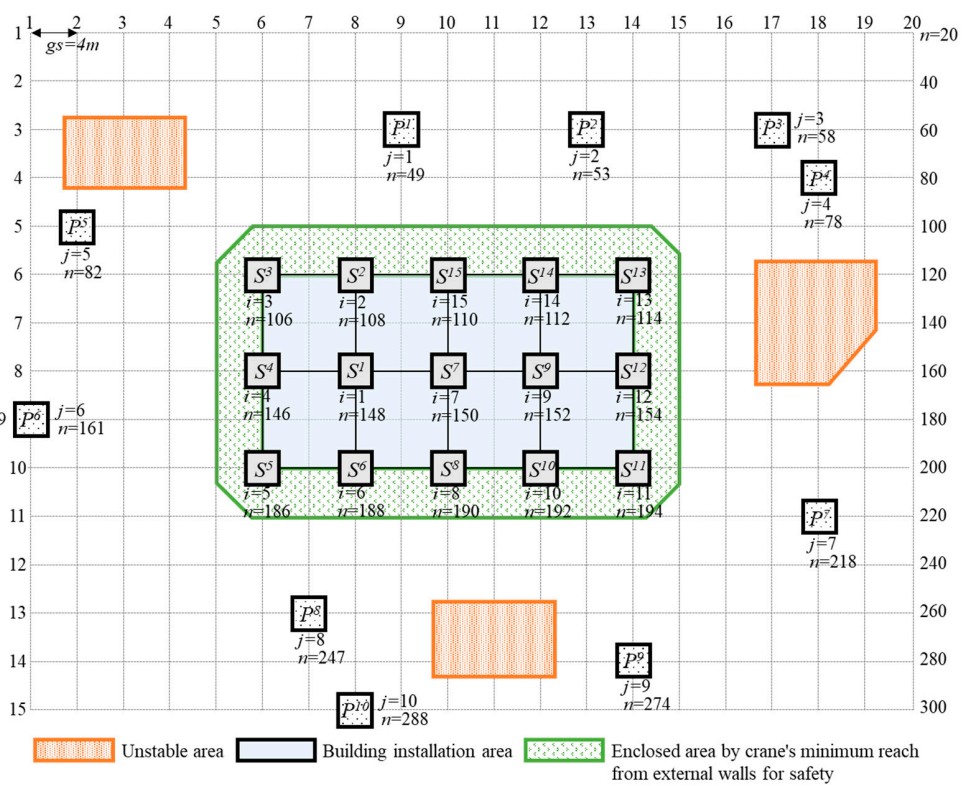

**Figure 4.** Case study site.

In the test, the grid size (*gs*) was set to be 4 m. The test site was converted into a grid according to the set grid size to create a 20 × 15 orthogonal coordinate system. The test site was represented using 300 points, and the point index n followed the linear index arrangement of the matrix. Therefore, according to the installation sequence, the set of column installation points (demand points) were [148, 108, 106, 146, 186, 188, 150, 190, 152, 192, 154, 194, 114, 112, 110], and the set of loading points (supply points) were [49, 53, 57, 78, 82, 161, 218, 247, 254, 288]. The set of points where it was impossible to move and place the mobile crane were [42, 43, 44, 45, 62, 63, 64, 65, 117, 118, 119, 137, 138, 157, 250, 251, 252, 290, 291, 292, 106:114, 126:134, 146:154, 166:174, 186:194, 85:20:205, 95:20:215, 86:94, 206:214],

a total of 90 points, including the building-occupied points, soft ground points, loading points, and points where the safe work distance was not secured.

*4.2. Optimization*

Table A1 in Appendix A shows the results obtained using Equations (6) and (11) to identify the feasible supply points ($E^{i,k}$) and the feasible points for the mobile crane ($F^{i,k,f}$) for the demand points ($S_i$) in this test case. Therefore, based on Equation (12), the size of the optimal solution search range in this test case was $3.09878 \times 10^{21}$ (= (3 + 1 + 2 + 3 + 3 + 3) × (13 + 5 + 4 + 1) × (10 + 1 + 11 + 8) × (2 + 12 + 13 + 4 + 2) × (5 + 13 + 15 + 10) × (3 + 14 + 2 + 11) × (3 + 3 + 3 + 2 + 2) × (10 + 6 + 9) × (3 + 3 + 2 + 1 + 2 + 1 + 3) × (5 + 4 + 12 + 4) × (1 + 4 + 6 + 6 + 10 + 5) × (1 + 1 + 15 + 1 + 16) × (5 + 15 + 16 + 13 + 2) × (11 + 15 + 8 + 6) × (14 + 11 + 1)).

The GA parameters were initialized with [*PS* = 400, *CR* = 0.4, *MR* = 0.05, StallGenLimit = 200 times, TimeLimit = Inf, StallTimeLimit = Inf] to execute the GA optimization operation. The GA operation stopped as the allowable cumulative change of the objective function reached $1 \times 10^{-6}$ after undergoing many generations.

RPOS found [(3, 1), (3, 4), (3, 9), (3, 4), (2, 1), (2, 8), (3, 3), (1, 5), (7, 1), (3, 4), (5, 3), (3, 2), (2, 4), (2, 6), (1, 7)] as the reposition route that minimized the operating time of the mobile crane (Table 1). These results were expressed using the chromosomes designed for the GA operation. Converting these results into the location points of the mobile crane yields [124, 124, 124, 124, 124, 231, 231, 231, 231, 231, 176, 176, 52, 52, 52]. In other words, the mobile crane operates by moving in a sequence of 124→231→176→52. It is shown that three relocations were required for the task. Furthermore, transforming the optimization operation results of the GA into pairs of the supply points for the demand points yields [82, 82, 82, 161, 161, 247, 247, 247, 254, 254, 218, 218, 53, 53, 49]. If these points are interpreted using the set of the column installation points (demand points), for which the installation sequence has been determined, and the placement sequence of the crane, the following results can be obtained. First, the mobile crane is positioned at point 124 to lift the columns to the demand points 148, 108, 106, 146, and 186. Here, the columns for demand points 148, 108, and 106 are fetched from the supply point 82, whereas the columns for demand points 146 and 186 are fetched from the supply point 161. Then, the crane moves to point 231 and lifts the columns to the demand points 188, 150, 190, 152, and 192. The columns for demand points 188, 150, and 190 are fetched from the loading point 247, whereas the columns for the demand points 152 and 192 are fetched from the supply point 254. Then, the crane moves to points 176 and 52 to continue loading (Figure 5).

**Table 1.** Experimental results.

| *i* | 1 | 2 | 3 | 4 | 5 | 6 | 7 | 8 | 9 | 10 | 11 | 12 | 13 | 14 | 15 |
|---|---|---|---|---|---|---|---|---|---|---|---|---|---|---|---|
| $S^i$ | 148 | 108 | 106 | 146 | 186 | 188 | 150 | 190 | 152 | 192 | 154 | 194 | 114 | 112 | 110 |
| Solution (*k,f*) | (3,1) | (3,4) | (3,9) | (3,4) | (2,1) | (2,8) | (3,3) | (1,5) | (7,1) | (3,4) | (5,3) | (3,2) | (2,4) | (2,6) | (1,7) |
| $E^{i,k}$ | 82 | 82 | 82 | 161 | 161 | 247 | 247 | 247 | 274 | 274 | 218 | 218 | 53 | 53 | 49 |
| $F^{i,k,f}$ | 124 | 124 | 124 | 124 | 124 | 231 | 231 | 231 | 231 | 231 | 176 | 176 | 52 | 52 | 52 |
| Lift time (*lt*) (s) | 205.2 | 190.9 | 174.0 | 189.6 | 145.8 | 91.2 | 145.8 | 139.8 | 155.8 | 140.2 | 244.2 | 174.0 | 164.1 | 155.4 | 106.4 |
| Travel time (*tt*) (s) | - | - | - | - | - | 44 | - | - | - | - | 28 | - | - | 32 | - |
| Reposit. time (*rt*) (s) | - | - | - | - | - | 600 | - | - | - | - | 600 | - | - | 600 | - |
| Total time (s) | | | | | | | 4326.4 | | | | | | | | |

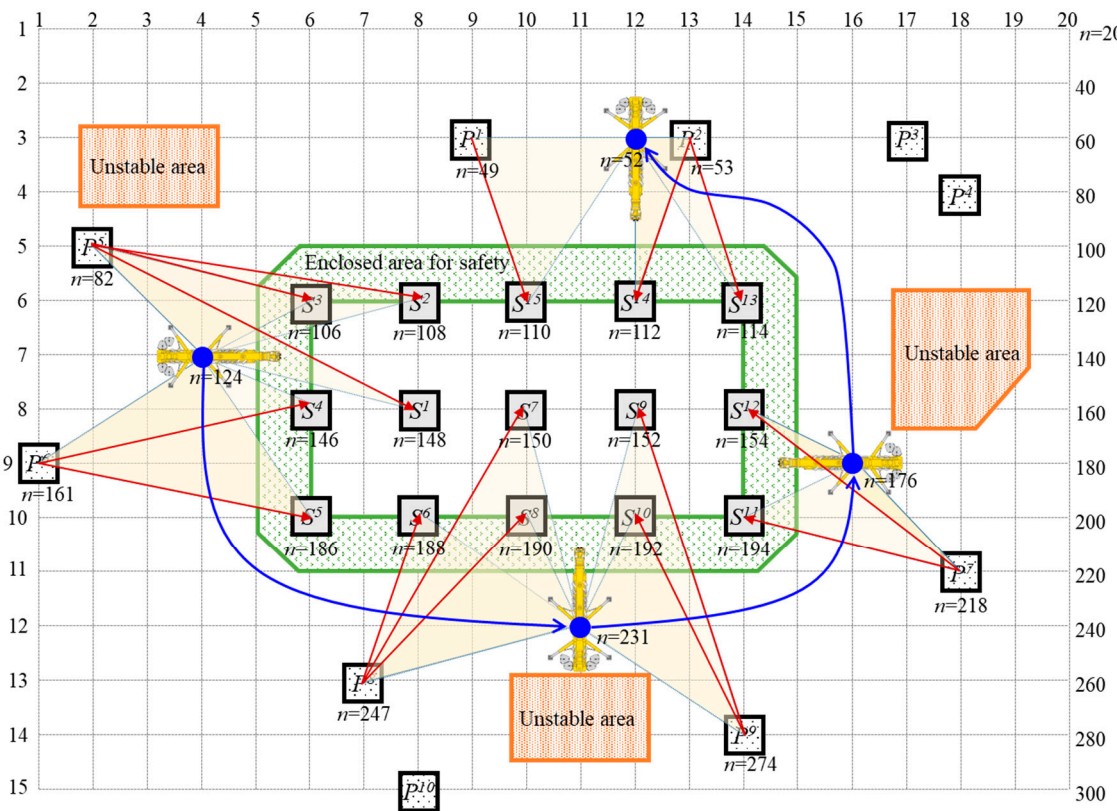

**Figure 5.** Graphical representation of the results.

The total required operating time of the mobile crane according to the GA operation result was 4326 s. However, this result does not consider the time required for hanging a column on the hook of the crane, the waiting time after lifting a column to the demand point, and the time before installing/fixing it after lifting. They are not included in the result because that time does not affect the optimal solution search. In the test case, as the column lifting work is performed 15 times, the total work time can be derived if the waiting time for installing/fixing is estimated and multiplied by 15 and then added to the required operating time of 4326 s. The decision-maker can check the result (Table 1, Figure 5) obtained by using our method to effectively and easily determine where the mobile crane should be located, where it should travel, and the location and payload quantity that should be loaded.

### 4.3. Comparison with Heuristic Method and Discussion

To validate the optimal solution search performance of RPOS, the RPOS result was compared to the results obtained by applying a conventional heuristic-based method to the same test case. The mobile crane reposition route suggested by the heuristic-based method was [124, 124, 124, 124, 124, 231, 231, 231, 231, 231, 136, 136, 136, 136, 50] ((i.e., 124→231→136→50), and the total required operating time was 4919 s. The mobile crane was relocated thrice, which was similar to the result of RPOS, but the total operating time required by this plan was 13.7% longer. This means that the optimal solution search performance of RPOS was higher. The search procedure of the heuristic-based method is as follows. First, the feasible points of the mobile crane for the *i*-th demand point were identified. Second, the feasible points for the *i*+1th demand point were identified. Third, coincident points between *i* and *i*+1th feasible points were identified. Fourth, the mobile crane was placed at a coincident point. Fifth, if the location point of the mobile crane belonged to the set of *i* + 2 feasible points, the crane remained at the current location point; otherwise, it moved to one of the *i* + 2 feasible points. This series of processes were

repeated for every demand point. The solution derived based on this approach minimized the relocation of the mobile crane. However, the optimal solution was not guaranteed. RPOS, on the other hand, found the near-global optimal solution that required less total operating time while minimizing the need to relocate the mobile crane. The difference in the solution search between the heuristic method and RPOS in this test case study was the decision made for the placement point of the mobile crane at the 11th demand point. The heuristic method decided to move the mobile crane to 136 because the point 136 was a feasible point for the 11th—14th demand points. In other words, the mobile crane was placed at a coincident point of four demand points. RPOS decided to move to 176, a feasible point for the 11th—13th demand points, which minimized the total required operating time. The difference in the total required operating time between the solutions suggested by the heuristic method and RPOS was 593 s, which may not be substantial in real world scenarios. However, this test case was performed at a relatively simple site. If there are more loads and the on-site situation is more complex, the difference in the optimal solution search performance will be much greater.

*4.4. Verifying the Computational Speed of RPOS*

In this chapter, the controlled experiment was conducted to verify the computational speed of RPOS. The experiment measures the convergence times taken to identify a near-global solution by using the same site and constraining variables used in test case 4.1, but changing the grid size (*gs*). The experimental results are shown in Table 2. In experiment 1 with a grid size of 4 m, RPOS identified the same near-global optimal solution that requires operating time 4326.4 s in 75.1 s, as shown in Table 2. When the grid size is decreased to 2 m (Experiment 2), the size of solution search spaces is $6.53481 \times 10^{31}$, which is much higher than for a grid size is 4 m ($3.09878 \times 10^{21}$). RPOS identified a better solution that requires operating time 4280.1 s in 384.61 s. The convergence time was increased by 5.1 times compared to the grid size of 4 m. When the grid size is decreased to 1 m (Experiment 3), the size of solution search spaces, convergence time, and operating time are $1.54221 \times 10^{42}$, 3738.91 s, and 4273.6, respectively. RPOS found a better solution, as the more the grid size decreases, the more the practicality of the mathematical models increases. On the other hand, the more the grid size decreases, the more the convergence time dramatically increases. As shown in the experiment, the user needs to consider the relationship between grid size, computational time, and total operation time, and determine an appropriate grid size according to the site conditions to be applied.

**Table 2.** Computational speed performance.

| Experiment Number | Grid Size (m) | Size of Search Spaces | Convergence Time (s) | Total Operation Time (s) |
|:---:|:---:|:---:|:---:|:---:|
| 1 | 4 | $3.09878 \times 10^{21}$ | 75.12 | 4326.4 |
| 2 | 2 | $6.53481 \times 10^{31}$ | 384.64 | 4280.1 |
| 3 | 1 | $1.54221 \times 10^{42}$ | 3738.91 | 4273.6 |

If it is necessary to reduce the computation time, it is recommended to apply A* instead of DA for finding the shortest paths between points because A* is known as being faster than DA. In addition, considering that the demand points are sequential, it can be recommended for clustering the demand points before executing GA so as to reduce the convergence time of RPOS."

## 5. Conclusions

This study proposes RPOS, which minimizes the total operating time required for the lifting work of a mobile crane. First, RPOS converts the construction site into a mathematical model using IFC file extraction, coordinate spacing setting, reference coordinate synchronization, geotechnical attribute inputs, etc. in the BIM model. Then, RPOS identifies the feasible locations for the mobile crane by considering the physical attributes of

the crane, attributes of the load, and safety regulations for lifting work. Furthermore, DA and GA-based optimization analysis methods are used to find the near-global optimal relocation route of the mobile crane. This method was developed as a MATLAB-based software, and the user can operate the system easily by inputting the aforementioned information (i.e., BIM model interlinkage, grid size, mobile crane attributes, payload attributes, and GA parameters). Furthermore, this system provides the user with visual information, such as (1) placement point sequences of the mobile crane, (2) demand point-supply point pairs, (3) travel route between the placement points of the mobile crane, and (4) total mobile crane operating time. This study provides, in detail, the computation process of the total operating time, which changes according to the placement of the mobile crane. Furthermore, the practical applicability was enhanced by suggesting a detailed mobile crane operating strategy reflecting the on-site situation. According to the results of the test case study, the near-global optimal solution search performance of RPOS is 13.7% better than that of a conventional heuristic-based mobile crane relocation search method. This study assumes that only one mobile crane is operated, and the mobile crane is limited to a common operation method, such as not moving while lifting the load. However, there are various mobile crane operation methods, such as moving while hoisting a load and collaboration between two cranes. In future studies, it is recommended to develop optimal crane operation planning methods incorporating various mobile crane operation methods.

**Author Contributions:** Conceptualization, methodology, and software, H.-S.G.; validation, H.-S.G. and H.-C.L.; formal analysis, investigation, H.-S.G., B.-Y.C. and Y.M.; writing—original draft preparation, H.-S.G.; writing—review and editing, H.-C.L.; visualization, B.-Y.C. and Y.M.; funding acquisition, H.-C.L. All authors have read and agreed to the published version of the manuscript.

**Funding:** This work was supported by the National Research Foundation of Korea (NRF) grant funded by the Korea government (MSIT) (No. NRF-2020R1C1C1013252).

**Institutional Review Board Statement:** Not applicable.

**Informed Consent Statement:** Not applicable.

**Conflicts of Interest:** The authors declare no conflict of interest.

## Notation

The following symbols are used in this article:

| | |
|---|---|
| $a_i$ | The binary number that determines whether to install/dismantle the crane |
| $A_C$ | The rotation angle of the crane |
| $b$ | The index of points occupied by building components ($B^b$) |
| $B^b$ | The points occupied by building components |
| $C_I$ | The table of mobile crane attributes data |
| $CR$ | The crossover rate |
| $D_{LI}$ | The distance between the supply point and demand point |
| $e$ | The index of the outer wall point ($E^e$) |
| $E_B$ | The set of the point ($E^e$) of outer walls |
| $E^e$ | The point of outer walls of the buildings among the points $B^b$ |
| $E^{i,k}$ | The available supply points for demand point ($S^i$) |
| $EF^{i,k,q}$ | The coincident points for the $S^i - E^{i,k}$ pair |
| $f$ | The index of the feasible point ($F^{i,k_f}$) for the $S^i - E^{i,k}$ pair |
| $f_{hoistt}$ | The lifting time estimation function |
| $f_{OT}$ | The fitness function |
| $F_L$ | The set of the feasible point ($F^{i,k_f}$) |
| $g$ | The index of the points ($SB^{e,g}$) where $R_C$ is not secured from each outer wall point |
| $gs$ | The grid size |
| $ht$ | The time of load lifting |
| $H_C$ | The height of the mobile crane body |
| $H_I$ | The height of the demand point |
| $H_R$ | The minimum rope length |

| | | |
|---|---|---|
| $i$ | The index of the demand point ($S^i$) | |
| $j$ | The index of the supply point ($P^j$) | |
| $k$ | The index of the available supply point ($E^{i,k}$) for the Si demand point | |
| $l$ | The number of loads to be installed | |
| $L_B$ | The maximum boom length | |
| $L_H$ | The horizontal length of the site area | |
| $L_M$ | The length of the payload | |
| $L_V$ | The vertical length of the site area | |
| $M$ | The number of rows of the grid | |
| $M_L$ | The matrix for storing points in the system, where the mobile crane cannot be installed | |
| $MR$ | The mutation rate | |
| $n$ | The index of the on-site grid coordinate point ($N_D{}^n$) | |
| $N$ | The number of columns of the grid | |
| $N_D{}^n$ | The point of on-site grid coordinate | |
| $P$ | The index of the points ($SF^{i,p}$) located within the range of $R_W$ centering on the $S^i$ | |
| $P^j$ | The supply points | |
| $P_L$ | The set of the supply points ($P^j$) | |
| $PS$ | The population size | |
| $q$ | The index of the coincident points ($EF^{i,k,q}$) for the $S^i - E^{i,k}$ pair | |
| $r$ | The index of the point where bearing capacity is not secured ($U^r$) | |
| $rt$ | The time of crane installing/dismantling | |
| $R_C$ | The minimum radius for safe lifting | |
| $RF^{i,k,t}$ | The points located within the range of $R_W$ centering on the $E^{i,k}$ | |
| $R_W$ | The maximum radius of the mobile crane | |
| $S^i$ | The demand point ($S^i$) | |
| $S_B{}^{e,g}$ | The points where $R_C$ is not secured from each outer wall point ($E^e$) | |
| $SF^{i,p}$ | The points located within the range of $R_W$ centering on the $S^i$ | |
| $S_L$ | The set of the demand point ($S^i$) | |
| $S_S$ | The search range of the optimal solution | |
| $T$ | The index of the points ($RF^{i,k,t}$) located within the range of RW centering on the $E^{i,k}$ | |
| $Tt$ | The time of the crane traveling | |
| $U^r$ | The point where bearing capacity is not secured | |
| $W_C$ | The weight of the mobile crane | |
| $W_M$ | The weight of the load | |

## Appendix A

Demand points, available supply points, feasible points for the mobile crane of case study are as follow:

**Table A1.** Demand points, available supply points, feasible points of case study.

| $i$ | Demand Point ($S^i$) | Available Supply Point ($E^{i,k}$) | Feasible Point ($F^{i,k,f}$) | Number of $F^{i,k,f}$ |
|---|---|---|---|---|
| 1 | 148 | 49 | 67, 68, 69 | 3 |
| | | 53 | 69 | 1 |
| | | 82 | 124, 144 | 2 |
| | | 161 | 124, 144, 164 | 3 |
| | | 247 | 227, 228, 229 | 3 |
| | | 288 | 227, 228, 229 | 3 |
| 2 | 108 | 15 | 27, 28, 29, 46, 47, 48, 50, 66, 67, 68, 69, 70, 71 | 13 |
| | | 53 | 29, 50, 69, 70, 71 | 5 |
| | | 82 | 66, 84, 104, 124 | 4 |
| | | 161 | 124 | 1 |

**Table A1.** *Cont.*

| $i$ | Demand Point ($S^i$) | Available Supply Point ($E^{i,k}$) | Feasible Point ($F^{i,k,f}$) | Number of $F^{i,k,f}$ |
|---|---|---|---|---|
| 3 | 106 | 15 | 25, 26, 27, 46, 47, 48, 66, 67, 68, 69 | 10 |
| | | 53 | 69 | 1 |
| | | 82 | 66, 83, 84, 102, 103, 104, 122, 123, 124, 143, 144 | 11 |
| | | 161 | 102, 103, 122, 123, 124, 143, 144, 164 | 8 |
| 4 | 146 | 49 | 66, 67 | 2 |
| | | 82 | 66, 84, 103, 104, 122, 123, 124, 142, 143, 144, 162, 163 | 12 |
| | | 161 | 103, 122, 123, 124, 142, 143, 144, 162, 163, 164, 183, 184, 204 | 13 |
| | | 247 | 204, 225, 226, 227 | 4 |
| | | 288 | 226, 227 | 2 |
| 5 | 186 | 82 | 124, 143, 144, 162, 163 | 5 |
| | | 161 | 124, 143, 144, 162, 163, 164, 182, 183, 184, 202, 203, 204, 223 | 13 |
| | | 247 | 204, 223, 224, 225, 226, 227, 228, 229, 244, 245, 246, 248, 265, 266, 267 | 15 |
| | | 288 | 226, 227, 228, 229, 245, 246, 248, 265, 266, 267 | 10 |
| 6 | 188 | 161 | 164, 184, 204 | 3 |
| | | 247 | 204, 225, 226, 227, 228, 229, 230, 231, 246, 248, 249, 267, 268, 269 | 14 |
| | | 254 | 230, 231 | 2 |
| | | 288 | 226, 227, 228, 229, 230, 246, 248, 249, 267, 268, 269 | 11 |
| 7 | 150 | 49 | 69, 70, 71 | 3 |
| | | 53 | 69, 70, 71 | 3 |
| | | 247 | 229, 230, 231 | 3 |
| | | 254 | 230, 231 | 2 |
| | | 288 | 229, 230 | 2 |
| 8 | 190 | 247 | 227, 228, 229, 230, 231, 248, 249, 269, 270, 271 | 10 |
| | | 254 | 230, 231, 232, 233, 270, 271 | 6 |
| | | 288 | 227, 228, 229, 230, 248, 249, 269, 270, 271 | 9 |
| 9 | 152 | 49 | 71, 72, 73 | 3 |
| | | 53 | 71, 72, 73 | 3 |
| | | 57 | 73, 136 | 2 |
| | | 78 | 136 | 1 |
| | | 218 | 156, 176 | 2 |
| | | 247 | 231 | 1 |
| | | 254 | 231, 232, 233 | 3 |
| 10 | 192 | 218 | 176, 196, 216, 234, 235 | 5 |
| | | 247 | 229, 230, 231, 271 | 4 |
| | | 254 | 196, 216, 230, 231, 232, 233, 234, 235, 253, 271, 272, 273 | 12 |
| | | 288 | 229, 230, 271, 272 | 4 |
| 11 | 154 | 49 | 73 | 1 |
| | | 53 | 73, 74, 75, 96 | 4 |
| | | 57 | 73, 74, 75, 96, 116, 136 | 6 |
| | | 78 | 74, 75, 96, 116, 136, 158 | 6 |
| | | 218 | 156, 158, 176, 177, 178, 196, 197, 216, 234, 235 | 10 |
| | | 254 | 196, 216, 233, 234, 235 | 5 |

**Table A1.** *Cont.*

| *i* | Demand Point ($S^i$) | Available Supply Point ($E^{i,k}$) | Feasible Point ($F^{i,k,f}$) | Number of $F^{i,k,f}$ |
|---|---|---|---|---|
| 12 | 194 | 57 | 136 | 1 |
| | | 78 | 136 | 1 |
| | | 218 | 156, 176, 177, 178, 196, 197, 198, 216, 217, 234, 235, 236, 237, 255, 256 | 15 |
| | | 247 | 231 | 1 |
| | | 254 | 196, 216, 217, 231, 232, 233, 234, 235, 236, 237, 253, 255, 256, 273, 274, 275 | 16 |
| 13 | 114 | 49 | 33, 52, 71, 72, 73 | 5 |
| | | 53 | 33, 34, 35, 52, 54, 55, 56, 71, 72, 73, 74, 75, 76, 77, 96 | 15 |
| | | 57 | 33, 34, 35, 54, 55, 56, 73, 74, 75, 76, 77, 96, 97, 98, 116, 136 | 16 |
| | | 78 | 35, 54, 55, 56, 74, 75, 76, 77, 96, 97, 98, 116, 136 | 13 |
| | | 218 | 156, 176 | 2 |
| 14 | 112 | 49 | 31, 32, 33, 50, 51, 52, 69, 70, 71, 72, 73 | 11 |
| | | 53 | 31, 32, 33, 50, 51, 52, 54, 69, 70, 71, 72, 73, 74, 75, 96 | 15 |
| | | 57 | 33, 54, 73, 74, 75, 96, 116, 136 | 8 |
| | | 78 | 54, 74, 75, 96, 116, 136 | 6 |
| 15 | 110 | 49 | 29, 30, 31, 48, 50, 51, 52, 67, 68, 69, 70, 71, 72, 73 | 14 |
| | | 53 | 29, 30, 31, 50, 51, 52, 69, 70, 71, 72, 73 | 11 |
| | | 57 | 73 | 1 |

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
