# Peer review of "GA-Based Optimization Method for Mobile Crane Repositioning Route Planning"

_applsci, doi:10.3390/app11136010_

Round 1

Reviewer 1 Report

It is a very interesting subject in which I present the following considerations:
It is reasonably well written, with some minor errors. It needs a slight improvement in English.
The article is very well organized; the chapters are well explained. The introduction is well structured and has the necessary and current references on the subject.

The main contribution of this paper is to solve the repositioning mobile crane problem using a Genetic algorithm.

It would be interesting to have a small section explaining the choice of the genetic algorithm instead of other solutions that could also solve this type of problem, such as Simulated Annealing and Tabu Search.

Why did it use Dijkstra and not A*? The A* is faster than  Dijkstra, and with the correct heuristic, it maintains optimality.

Be careful with the use of the word Optimal.
Please note that the genetic algorithm does not guarantee to find the Optimal Mobile Crane Travel Route. I am an apologist for saying that GA finds a sub-optimal route.

It was interesting and important to find out how long it took to reach the solution. Moreover, it would be very interesting to know how much longer it took if the grid size was reduced to 1m. Grids with 4m mean that the solution can be further away from optimal than expected.

Minor error
In equation 1,  M is related to the vertical and N to the horizontal, and in figure 1, it is the opposite.

Reviewer 2 Report

The paper is focused on finding the optimal position/route of mobile cranes in construction sites. It is well written and technically sound. The problem formulation and the graphs are very clear. However, I have some suggestions that can improve the quality of the paper.

1.In section 3.3, the authors said the load installation is sequential. Therefore, for example in section 4, the optimal solution of the crane location at n=176 is independent from the crane location n=124. Or n=53 is independent from n=231,124. If this is true, the optimization problem can also be solved sequentially which will greatly reduce the computation cost. 

2. It seems that the feasible set of the problem is very large (even with a large grid size) and I guess the problem is highly nonlinear making it difficult to find an optimal solution. It would be nice to add a discussion about the computation time or even include the computation time for the example in section 4. 
